

# CLEAR: a new discrete multiplicative random cascade model for disaggregating path-integrated rainfall estimates from commercial microwave links

Martin Fencl[1], Marc Schleiss[2]

[1]Department of Hydraulics and Hydrology, Czech Technical University in Prague, Prague, Czech Republic
[2]Department of Geoscience and Remote Sensing, Faculty of Civil Engineering and Geosciences, Delft University of Technology, the Netherlands

*Correspondence to*: Martin Fencl (martin.fencl@cvut.cz)

**Abstract.** A novel disaggregation algorithm for commercial microwave links (CMLs), named CLEAR (CML Segments with

Equal Amounts of Rain), is proposed. CLEAR utilizes a multiplicative random cascade generator to control the splitting of link segments, with the generator's standard deviation dependent on the rain rate and segment length. Spatial consistency during the splitting process is maintained using rain rate information from neighboring CMLs. CLEAR is evaluated on a network of 77 CMLs in Prague. The performance is assessed first using simulated rainfall fields and second through a case study with real attenuation data from the network to demonstrate its applicability in real-world scenarios. Results from the

virtual rainfall fields indicate good overall performance, including the generation of realistic spatial patterns. CLEAR effectively estimates maximal and minimal rain rates along CML paths and outperforms a commonly used benchmark algorithm. The stochastic nature of CLEAR allows it to represent uncertainty as an ensemble of rain rate distributions along CML paths. However, the generated ensembles significantly underestimate overall variability along the paths. Additionally, the case study on real data highlights challenges associated with uncertainties in CML quantitative precipitation estimates, which are common across all methods. In conclusion, CLEAR contributes to generating more representative rainfall

distributions along CMLs, which is critical for spatial reconstruction of rainfall fields from path-integrated CML data. It also has the potential to reduce errors in CML quantitative precipitation estimates caused by assuming uniform rain rates along CML paths.

## 1 Introduction

Commercial microwave links (CMLs) are point-to-point radio connections in cellular networks. They typically operate at frequencies in the order of 10-90 GHz (Chwala and Kunstmann, 2019; Fencl et al., 2020) where electromagnetic waves are known to be attenuated by raindrops. This attenuation can be measured and used to provide path-averaged rainfall estimates (Leijnse et al., 2007; Messer et al., 2006).



CMLs are an appealing source of opportunistic rainfall measurements. According to (Ericsson, 2019), there are about 5 million
CMLs worldwide, including sparsely gauged regions and developing countries. The large coverage, high density in urban
areas, and low costs of operation are clear advantages over traditional rain gauge and radar networks. However, the path-
integrated nature of CML data also poses some challenges. For example, if one wishes to retrieve spatially representative
rainfall estimates (e.g., 2D maps), the path-integrated data from the CMLs first need to be transformed to point data and
interpolated to a regular two-dimensional Cartesian grid. The most straightforward way to obtain such a map is to reduce each
CML observation into a single-point measurement located at the center of the CML path and subsequently interpolate these
point data using kriging or inverse distance weighted (IDW) interpolation (Graf et al., 2019; Overeem et al., 2013).
Unfortunately, previous research has shown that due to the large spatial and temporal variability of rain, such an approach can
lead to large biases and unrealistic rainfall distributions, especially for longer CMLs of several kilometers in length and during
heavy, localized rain showers.

Over time, several alternative solutions to the rainfall reconstruction problem from CML data have been proposed.
Tomographic reconstruction methods (Cuccoli et al., 2013; Giuli et al., 1991; Zinevich et al., 2008) offer the advantage of
directly handling path-averaged rainfall data. Another approach, random mixing, achieves this by conditioning random fields
with a spatial dependence structure modelled by copulas (Haese et al., 2017). However, in both cases optimal performance
requires a model of the underlying rainfall field, which is often unavailable. Following a different approach, Goldshtein et al.,
(2009) suggested an iterative reconstruction algorithm based on IDW interpolation where each CML is represented by a set of
equally spaced points. The distribution the rainfall rates along the control points is then iteratively estimated from observations
of neighboring CMLs, until some kind of convergence is reached. Both tomographic and iterative IDW algorithms are
computationally efficient, with decent performances for slowly varying rainfall fields and a more or less regular network of
CMLs. However, their performance strongly depends on CML topology (e.g., link density, lengths, frequencies and
orientations) and rainfall variability. So far, no convincing solution has been proposed to address the issue of rainfall
intermittency (i.e., the fact that it may not rain over the entire CML), which is a big problem for longer CMLs and during
heavy, localized rain showers. In those cases, both tomographic and IDW-based algorithms are likely to predict highly
unrealistic spatial structures and distributions with large outliers and uncertainties. This paper addresses this issue by means
of a novel disaggregation technique based on random cascades named CLEAR (CML segments with equal amounts of rain).
CLEAR redistributes rainfall amounts along CML paths over smaller and smaller scales by means of a discrete, conservative
multiplicative random cascade. The approach is inspired by the EVA (Equal-volume area) cascade by (Schleiss, 2020) for
disaggregating spatially intermittent rainfall fields. During the cascade, each CML segment is split into two new segments of
different path-lengths but identical path-integrated rainfall. Random cascades have been extensively used to downscale time
series and spatial fields of rain. However, to our knowledge, this is the first time that the formalism is applied to path-averaged
data from CMLs.  Because CLEAR inherits the main features of the EVA cascade model, we believe that it is better suited to
reproduce the highly variable rainfall distributions seen along CMLs, including its intermittency. Furthermore, the stochastic





nature of the cascade makes it possible to quantify the uncertainty related to the spatial redistribution of rainfall rates along CMLs.

As with any random cascade model, the performance of CLEAR strongly depends on the characteristics of the underlying
cascade generator model. Hence, different ways to model and estimate the generator model based on high-resolution virtual rainfall fields are proposed and discussed. In addition to the simulation experiments, we also report on the results obtained for a case study in Prague (CZ), which we use to highlight the strengths and weaknesses of CLEAR compared with other approaches. For simplicity, the scope is limited to discussing how to disaggregate path-averaged CML rain rates using CLEAR while other important issues related to the spatial interpolation and final reconstruction of 2D rainfall fields from CML data
are ignored.

The rest of this paper is structured as follows: The method section describes the algorithm and virtual rainfall fields used for performance evaluation. Performance metrics and comparison to the benchmark by (Goldshtein et al., 2009) are presented in the Results section. The case study section demonstrates the strengths and weaknesses of the algorithm on real CML observations. Finally, the results and limitations of the algorithm are critically reviewed and contextualized in the discussion
and Conclusion sections.

## 2 Material and methods

All essential data and codes underlying this publication are openly available on 4TU Research data (Fencl and Schleiss, 2025).

### 2.1 Rainfall estimation from commercial microwave links

The basic quantity needed to estimate rainfall from CMLs is the total loss ($L_t$) in power between the transmitted and received
signals. The total loss consists of various components, including free space loss, losses in the medium (e.g. gaseous attenuation and raindrop attenuation), losses at transmission and reception, and antenna gains. Before the rainfall rate can be estimated, different types of signal processing techniques need to be applied to identify and separate the rainfall-related specific attenuation $k$ (in dB km$^{-1}$) from other sources of attenuation. We write:

$$k = max\left(\frac{L_t - B - A_w}{l}, 0\right),\tag{1}$$

where $l$ (km) is the length of the CML path, $B$ (dB) is the baseline attenuation (i.e., all losses that are not due to rain) and $Aw$ (dB) the wet antenna attenuation due to water on the antenna radomes. For an overview of different baseline and wet antenna attenuation estimation techniques, the reader is referred to (Chwala and Kunstmann, 2019; Pastorek et al., 2022a). Once the specific rainfall-induced attenuation has been retrieved, a power-law model can be used to estimate the path-averaged rainfall rate $R$ (mm h$^{-1}$) along the link (Atlas and Ulbrich, 1977):

$$R = \alpha k^{\beta},\tag{2}$$

where $\alpha$ and $\beta$ are empirical parameters dependent on CML frequency, polarization, and raindrop size distribution (ITU-R, 2005).



## 2.2 The CLEAR algorithm

The CLEAR algorithm redistributes the path-integrated rainfall amount along a CML over smaller and smaller scales by means

of discrete multiplicative random cascade. At each cascade level the CML segments are split into two smaller segments of variable lengths, containing half of the original rainfall amount. The ratio between the parent length $L_0$ and resulting segment lengths $L_1$ and $L_2$ is determined by drawing random weights $W_1$ and $W_2 = 1 - W_1$ from a cascade generator model with logit normal probability distribution:

$$ln\left(\frac{W}{1-W}\right) N(\mu = 0, \sigma),\qquad(3)$$

where $\mu$ is the mean and $\sigma$ the standard deviation of an underlying Gaussian random variable. The mean $\mu$ is forced to zero, to ensure $W$ is centered around 0.5. The path-averaged rain rates $R_1$ and $R_2$ along the two split segments satisfy the following relations:

$$L_0 W_1 R_1 = L_0 W_2 R_2 = \frac{1}{2}L_0 R_0,\qquad(4)$$

where $R_0$ (mm h$^{-1}$) is the path-averaged rain rate of the parent CML segment and $W_1$ and $W_2 = 1 - W_1$ are the random cascade

weights.

The splitting can be controlled by changing the standard deviation of the generator (3). For small standard deviation values, the random weights tend to be closer to 0.5, which leads to a more homogeneous redistribution of the rainfall rates along the path of the link. For larger values of standard deviation, the weights cluster around 0 and 1, which translates into more uneven splits and more intermittency (Schleiss, 2020).

During the splitting process, a spatial coherence rule is applied to determine which link segment receives the shortest length and, therefore, the highest rainfall intensity along its path. According to this rule, the smaller of the two weights (W1, W2) is always assigned to the link segment experiencing the highest rainfall rate in its vicinity, based on neighboring segments that have already been split. This approach works under the assumption that all CML segments are split only once at the first cascade level before progressing to the next level.

To estimate the rainfall rate in the vicinity, the spatial coherence rule involves an intermediate step: a partial spatial reconstruction of the rainfall field over a regular Cartesian grid. For further details on this process, readers are referred to Appendix A.

The splitting process concludes when the path-integrated rainfall along a CML segment falls below a predefined threshold. CML segments with very small rainfall amounts are no longer split but continue to be considered when applying the spatial

coherence rule to the remaining CML segments. The cascade terminates when all CML segments have stopped splitting or when a fixed number of cascade levels has been reached. Since the cascade weights are drawn at random, the CLEAR algorithm produces a different output each time it is run. By comparing the different realizations to each other, one can quantify the uncertainty (in point rainfall estimates) due to the random redistribution of the rainfall rates along the CMLs. For technical details of the CLEAR implementation, readers are referred to scripts "fun_CLEAR.R" and "04_CLEAR_disaggregation.Rmd"

(Fencl and Schleiss, 2025).



### 2.3 Approach

CLEAR is tested on real CML data from a telecommunication backhaul operated by T-Mobile, CZ in Prague. First, simulated rainfall fields with 6000 randomly placed CMLs of various lengths between 0.5 and 6 km are used to calibrate the cascade generator, for implementation details readers are referred to a script named "01_sample_estimation_empirical_weights.Rmd" (Fencl and Schleiss, 2025). Then the performance of CLEAR is assessed with respect to the GMZ method by Goldshtein et al., (2009) using simulated rain rates with real-world topology of 77 CMLs over a domain of size 20 x 20 km$^2$ in Prague (Fig. 1). Finally, real attenuation data from the same set of 77 CMLs are used as a case study to illustrate the strengths and weaknesses of the approach in real-world applications.

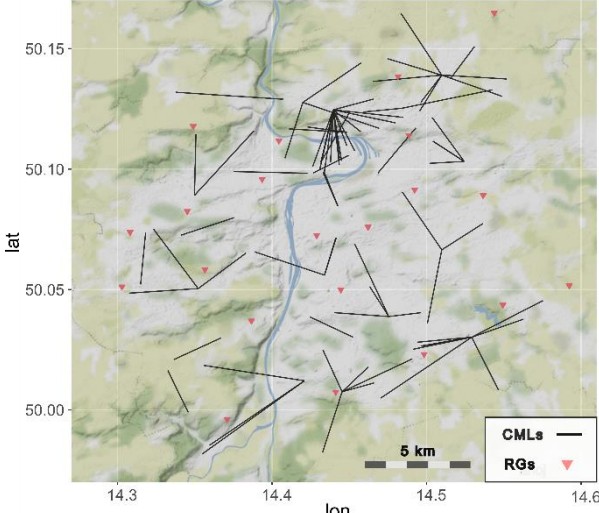

**Figure 1: Map of CMLs in Prague used for the analysis, together with the rain gauges used for bias-correcting weather radar rainfall estimates over the area. ©OpenStreetMap Distributed under the Open Data Commons Open Database License (ODbL) v1.0.**

### 2.4 CML data

The CML dataset used in the case study was acquired during the month of September 2014 by an SNMP based application running at T-Mobile network operation centre. The data consist of CML transmit and received signal power levels recorded at approximately 10-second intervals. The selected CMLs operate at frequencies between 23 and 38 GHz, and only those longer than 1.5 km were included in the analysis, resulting in a total of 77 CMLs. The 1.5 km length threshold was chosen to align with the 1 km² resolution of the weather radar reference used in the case study. Additionally, shorter CMLs in real networks are significantly affected by errors related to wet antenna attenuation and quantization, making them less suitable for rainfall retrieval (Blettner et al., 2023).



## 2.5 Simulated rainfall fields

Virtual rainfall fields were generated to represent three events with varying intensities, durations, and spatial variabilities (see Table 1), following the method proposed by (Schleiss et al., 2012). These simulated fields cover a 20 x 20 km² area, with a spatial resolution of 100 x 100 m² and a temporal resolution of 1 minute. The fields are advected and evolved over time in a prescribed direction to create a realistic spatio-temporal structure. Rainfall rates extracted from these simulated fields serve as reference values for evaluating the performance of the CLEAR algorithm (Sect. 2.8). The simulated rainfall fields are provided as a zipped csv file named "rainfall_field_rain_rates.csv" (Fencl and Schleiss 2025).

**Table 1: Characteristics of the virtual rainfall events. The metrics are calculated for a whole domain for each time step.**

|  | Duration (min) | Max. rain rates (mm h⁻¹) | Mean rain rates (mm h⁻¹) | Intermittency (%) | Advection direction |
|---|---|---|---|---|---|
| Event 1 | 30 | $41.8 - 51.3$ | $6.6 - 9.2$ | $0.1 - 6.7$ | NW → SE |
| Event 2 | 120 | $18.4 - 32.1$ | $1.9 - 9.7$ | $0.0 - 19.5$ | W → E |
| Event 3 | 60 | $0.3 - 30.5$ | $0.1 - 1.9$ | $35.2 - 99.5$ | NW → SE |

## 2.6 Radar rainfall for the case study

Bias-adjusted C-band weather radar rainfall estimates provided by the Czech Meteorological Institute are used as a reference rainfall when validating the results for the case studies. Specifically, we used the gridded product from the lowest elevation layer (Cappi2000) which has a spatial resolution of 1 x 1 km² and a temporal resolution of 5 minutes. The rainfall estimates were adjusted using the mean field bias correction method in *wradlib*, an Open Source Library for Weather Radar Data Processing (Heistermann et al., 2013). Note that the mean field bias was estimated using 23 tipping bucket rain gauges of type MR3, METEOSERVIS v.o.s. (operated by the city of Prague), with a catch area of 500 cm² and a tip resolution of 0.1 mm.

## 2.7 Sample estimation of the cascade generator model

Similarly to the original EVA cascade in Schleiss (2020), the standard deviation (SD) of the cascade generator is assumed to depend on the length $L_0$ (km) of the parent CML segment and rain rate $R_0$ (mm/h) according to the following power-law model:

$$SD = aL_0^b R_0^c, \tag{5}$$

where, $a$ (km⁻ᵇ mm⁻ᶜ hᶜ), $b$ (-), and $c$ (-) are empirical parameters that need to be estimated from the data or prescribed by the user.

There are two ways to estimate the cascade generator model: (1) using real data, and (2) using simulated rainfall fields. The first approach is purely data-driven. Given a set of CMLs with varying lengths, orientations, and path-integrated rainfall intensities, the key question is: how should a CML be split to ensure that the resulting segments have the same path-integrated rainfall attenuation (or, equivalently, the same total rainfall amount)? The answer depends on many factors such as the link's




length, position, orientation, and the characteristics of the rainfall field. This is why multiple CMLs and rainfall fields are needed to estimate a robust, climatological cascade generators.

However, estimating empirical cascade weights using real CML networks and gridded weather radar data has drawbacks.
Radar products often lack the spatial resolution needed to accurately capture rainfall variability along CMLs, particularly for shorter links. Additionally, results may be highly specific to the particular CML network, which can be problematic if link lengths and orientations are not equally well represented. Furthermore, measurement noise in both radar and CML data complicates the estimation process, making it challenging to obtain precise estimates of empirical cascade weights. The simulation approach addresses these issues. By using large synthetic CML networks with diverse lengths and orientations,
along with high-resolution simulated rainfall fields that realistically represent the local climatology, one can more accurately estimate the empirical cascade generator model.

For an arbitrary CML of length $l$ and rainfall field $R(x)$, the empirical breakdown coefficients $w$ can be calculated by splitting the CML such that:

$$\frac{1}{2}\int_0^l R(x)dx = \int_0^{wl} R(x)dx, \tag{6}$$

Simulated rainfall fields have much higher spatial resolutions than radar. Nevertheless, there will always be some discretization level, which means that in practice, the integral in (6) has to be replaced by a cumulative sum. The exact position of the breakpoint $W$ is thus determined by linear interpolation (Fig. 2). The breakdown coefficients $W$ are then transformed using the left-hand side of (3) to follow Gaussian distribution and grouped according to the path lengths and path-averaged rain rates of the parent links that generated them. The sample SD is then calculated for each group which empirically relates SD with rain-
rate and path-length (Fig. 3, left). SD model (5) is then optimized to fit empirically estimated SD (Fig. 3, middle, right). For the implementation of the optimization procedure, we refer reader a R Notebook named " 02_sample_estimation_fitting_SDmodel.Rmd" (Fencl and Schleiss 2025). The optimal parameters for our case are $a$ = 0.65, $b$ = 0.28, and $c$ = -0.33. With these parameters, SD tends to be high especially for very low rain rates and long CMLs, leading to the high probability of unequal splits. Conversely, at higher rain rates and for shorter CMLs, the splits are more likely to
approach an even division. More details on how to calculate the sample SD and fit the SD model (5) are given in Appendix B.

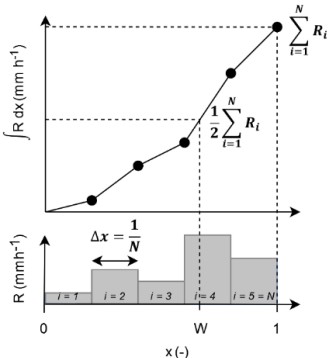

**Figure 2: Empirical breakdown coefficient determined by a cumulative some of rain rates along a CML path using linear interpolation.**





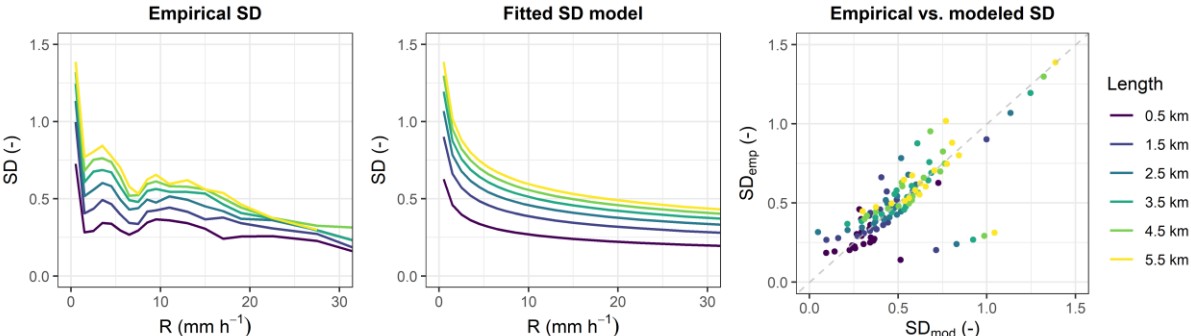

**Figure 3: Left: Standard deviation calculated for all samples larger than 50 members. Middle: Fitted SD model. Right: comparison of empirical and modeled standard deviations. Path lengths of parent CMLs are distinguished by color.**

## 2.8 Benchmarking and performance evaluation

Path-averaged rain rates without disaggregation are used as a zero benchmark algorithm. The disaggregation procedure implemented in the GMZ algorithm (Goldshtein et al., 2009) is used as a more complex benchmark: Each CML is divided into segments of equal length such that the length is close to some predefined threshold (100 m in our case) but does not exceed it. This threshold is the same for all the CMLs in the domain and determines the final resolution of the disaggregation. The resulting CML segments have lengths between 94 and 100 m. Initial path-averaged rain rates are iteratively redistributed along CML segments to match rain rates interpolated from neighboring CMLs with inverse-distance-weighted mean.

The performance of CLEAR is assessed in detail using synthetic experiments with virtual rainfall fields. High-resolution rainfall fields placed over the network of CMLs enables us to extract reference distributed rain rates and path-averaged rain rates along the path of each CML. Path-averaged rain rates are in each time step disaggregated with CLEAR algorithm and compared to the reference rainfall. Furthermore, the disaggregation performance is benchmarked against GMZ algorithm. To enable benchmarking, the reference and CLEAR-disaggregated rain rates are resampled using weighted average to match the segments defined by GMZ algorithm.

Three different features of disaggregation algorithms are evaluated:

1. Ability to reproduce rainfall patterns and extremes along a CML path is evaluated by quantifying standard deviation of rain rates, their maxima, and their minima along each CML path during each time step. In addition, we quantify variance conditional to rain rate and CML length.

2. Disaggregated rain rates are directly compared to the reference, i.e. besides the distribution also the location of rain rate estimates are considered.

3. An ensemble of CLEAR rain rates (50 runs) is generated and evaluated in terms of its variance.

R-squared ($R^2$), root mean square error ($RMSE$), and relative error ($RE$) are used as performance metrics in the first and second analysis. Containing ratio ($CR$) and average band width ($ABW$) are used as a performance metrics in the third analysis. $CR$ is





defined as the ratio of observations lying within confidence bands defined by 5 % a 95 % quantile of the whole ensemble and *ABW* as an average difference between 5 % a 95 % quantile of the whole ensemble.

*CR* and *ABW* is evaluated for different classes of rain rate and CML length. The same classes are used when quantifying conditional variance in the first analysis. Five equidistant CML length classes are defined covering lengths 1 – 6 km. Rain rate classes are defined by non-equidistant binning along the range of rain rates available in the dataset (0 - 52 mm h$^{-1}$): Rain rates 0 - 10 mm h$^{-1}$ are binned by 1 mm h$^{-1}$, binning by 2 mm h$^{-1}$ is applied up to rain rate of 20 mm h$^{-1}$, binning by 5 mm h$^{-1}$ up to 30 mm h$^{-1}$, and final two classes are 30 – 40 mm h$^{-1}$ and 40 – 52 mm h$^{-1}$. Relatively large size of bins for high rain rates reflects their low number in the dataset (and in general).

## 3 Results

The results in the following subsections are obtained from the experiment with simulated rainfall fields (Sect. 2.5). The spatial resolution of disaggregated and reference rain rates is 100 m.

### 3.1 Features of CLEAR disaggregation

Figure 4 shows two examples of rainfall rates disaggregated with the CLEAR algorithm. In the first case (Fig. 4, left), CLEAR nicely reproduces the actual distribution of rain rates along the CML. The location of the min/max values are estimated correctly, the estimated ensemble mean is moderately correlated with the reference rain rates (r = 0.55), and the variance over the ensemble members nicely captures the overall variability of the rainfall rate along the link (i.e., 89 % of the reference observations lie within the 90% confidence bands). Moreover, the ensemble spread tends to increase with growing rain rates (r = 0.51), reflecting higher uncertainty due to disaggregation during heavy rainfall. In the second case (Fig. 4, right), while the overall variability in rainfall rates along the link is accurately captured, the locations of the predicted minima and maxima are incorrect, and the 90% confidence bands do not align with the actual observations. This highlights an important point: in CLEAR, the position of peak rainfall intensity along a CML is heavily influenced by the spatial distribution of rainfall in the surrounding area. When no nearby CMLs are available, the spatial consistency rule relies almost entirely on smooth spatial interpolation of the rainfall field at coarser levels (see Appendix A). As a result, for isolated CMLs, CLEAR tends to systematically assign peak rainfall intensity to the same side of the link.

The better performance observed for CML 63 can likely be attributed to the presence of nearby CMLs, which provide valuable information to the spatial consistency rule and significantly influence the splitting process and the location of the peak intensity. In contrast, CML 9 is near the domain border, with only one end point having independent CML observations in its vicinity, which limits the accuracy of the prediction.



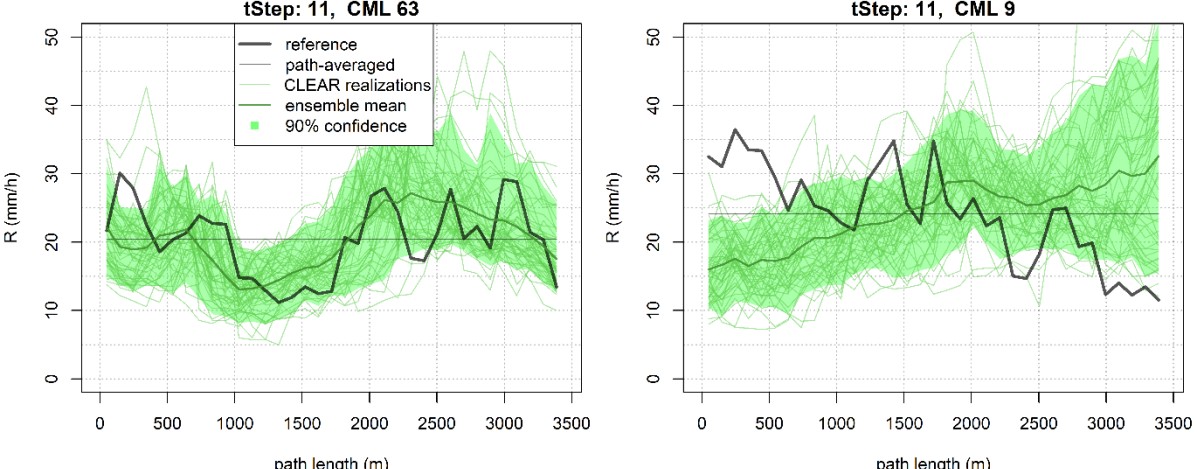

**Figure 4: 50 realizations of CLEAR rain-rate disaggregation along two links (CML 63 and CML 9) of similar path length (i.e., 3.5 km) during the time step 11 of event 1. Ninety-percent confidence bands are calculated as 5 % and 95 % quantiles of all realizations over each CML segment. Reference rain rates are from simulated rainfall fields.**

## 3.2 Evaluation of rainfall patterns along CMLs

The ability of disaggregation algorithms to realistically reproduce rainfall patterns is evaluated in each time step by quantifying rain-rate maximum, minimum, and standard deviation along a path of each CML. Figure 5 compares the statistics for the reference rain-rates with the ones obtained using CLEAR disaggregation and the two benchmark algorithms. The ensemble of 50 CLEAR realizations is treated in two different ways: a) the statistics are evaluated for a single realization and b) the statistics are evaluated for each realization and afterwards averaged. The CLEAR algorithm clearly outperforms the benchmarks in all three statistics. It is better at reproducing the min/max values (overall, across all ensemble members as well as for individual realizations). The ensemble mean of the statistics leads to even more robust results. Relative error is similarly low as by the single realization, nevertheless, *RMSE* is markedly lower and $R^2$ higher. Figure 5 also clearly shows how the naive approach of taking path-averaged rain rates systematically underestimates local maxima and overestimates minima.





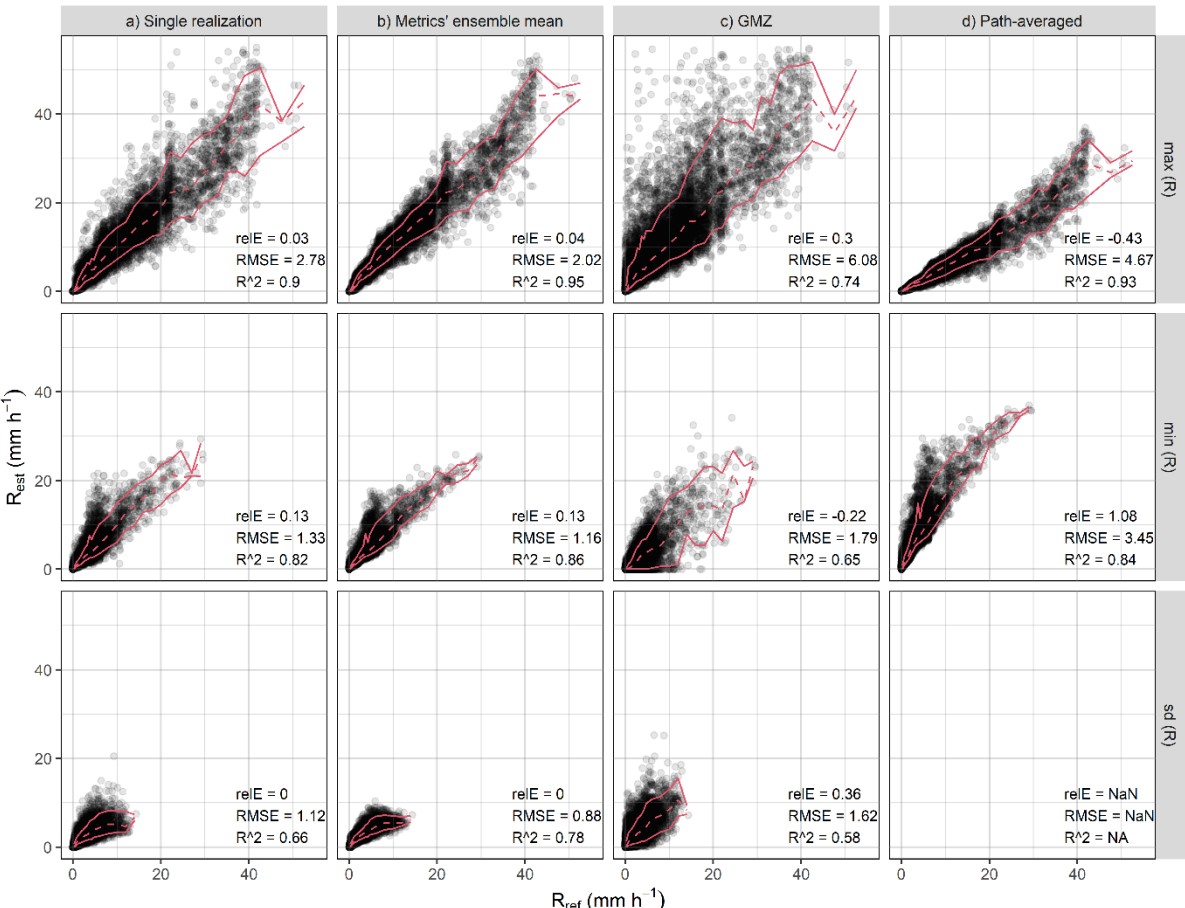

**Figure 5: Statistics of reference rain rate ($R_{ref}$) along a CML path quantified for each time step compared to the statistics of estimated rain rates ($R_{est}$) when using CLEAR algorithm (a)-(b), benchmark GMZ algorithm (c), or path-averaged rain rates without disaggregation (d). Red lines depict median and 10% and 90% quantiles.**

CLEAR also reliably accounts for the effect of rain rate averaging along a CML path. Figure 6a shows how the variance of reference rain rates along a CML on average increases with increasing rain rate. In addition, for low and moderate rain rates, the variance tends to be higher for longer CMLs. CLEAR is able reproduce the dependence of variance on both rain rate and CML length very well up to rain rates about 15 mm h$^{-1}$. For higher rain rates, the variance tends to be overestimated. This is probably due to systematic overestimation of the SD model (5) during higher rain rates (Fig. 3). The GMZ algorithm tends to overestimate variance and fails to accurately capture the relationship between variance and CML path length. Specifically, it does not reflect that variance increases with longer CML path.

Compared with GMZ, CLEAR also has a more stable performance: Figure 7 shows R-squared between reference and disaggregated rain rates along each CML evaluated over all time steps. R-squared values for the CLEAR ensemble mean range between 0.69 and 0.93 with a median value of 0.80, while for GMZ the values are between 0.3 and 0.94 with a median of 0.68. For comparison, the R-squared values for the zero benchmark (path-averaged rain rates) are between 0.63 and 0.94 with a





median of 0.83. By CLEAR 18 CMLs (23 %) perform better than zero benchmark, whereas by GMZ it only 6 CMLs (8 %) perform better. It is interesting, that none of the CMLs performing better with GMZ match with those performing better under

CLEAR disaggregation, which shows how CLEAR can help overcome the weaknesses of GMZ.

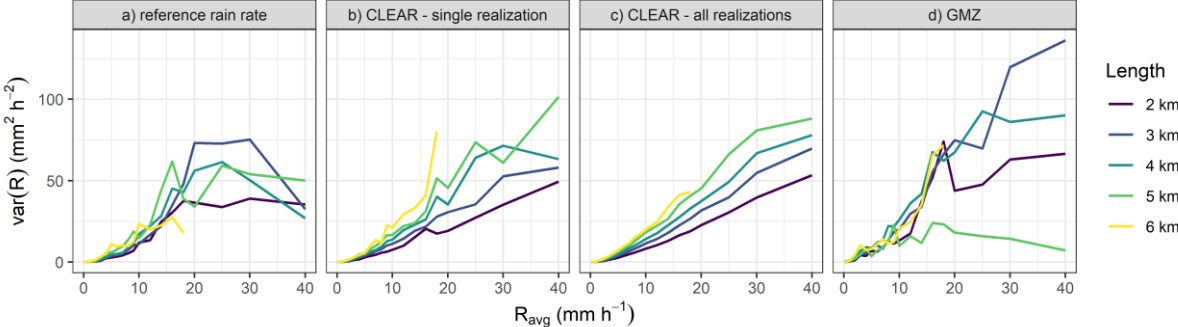

**Figure 6: variance along a CML path conditional to rain rate and CML length**

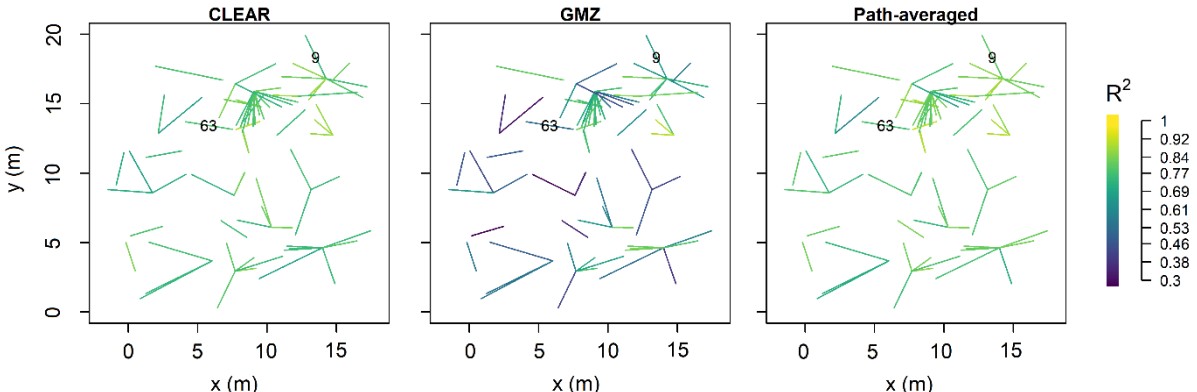

**Figure 7: R-squared between reference and disaggregated rain rates evaluated over all time steps separately for each CML. Left: CLEAR ensemble mean. Middle: GMZ. Right: GMZ.**

### 3.3 Segment-by-segment evaluation

The experiments performed on the simulated rainfall fields show that CLEAR produces roughly unbiased estimates on average. However, a more detailed segment-by-segment comparison between the ensemble mean of CLEAR (Fig. 8, left) and the actual

rainfall values shows a clear conditional bias as a function of rainfall intensity.





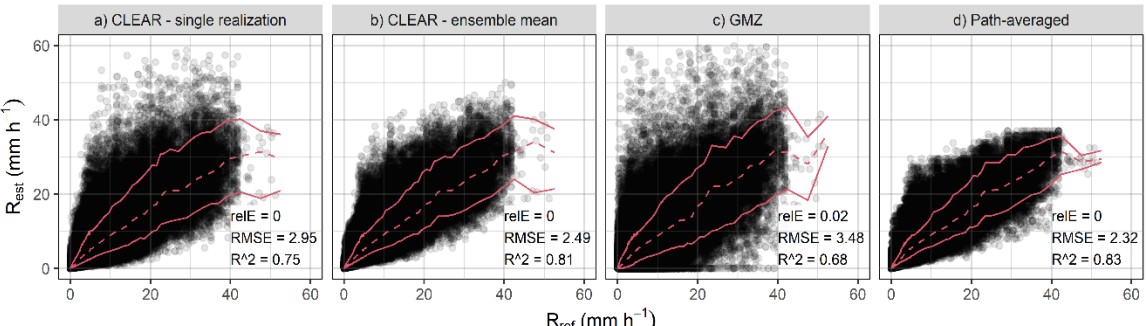

**Figure 8: Segment-by-segment comparison of disaggregated (a, b, c) and path-averaged (d) rain rates to the reference with lines depicting median and 10% and 90% quantiles.**

Figure 8 shows that CLEAR systematically overestimates low rainfall rates and underestimates higher ones. This conditional

bias can be attributed to errors in location with respect to the min/max rainfall rates along the link segments, as demonstrated in the right panel of Fig. 4, as well as the way the link segments are split during the cascade. However, it should be noted that the GMZ benchmark suffers from the same conditional bias. Moreover, GMZ also produces more outliers, even during relatively low rainfall rates. CLEAR does not have this issue because the disaggregation is controlled by a rain-rate dependent generator model, which means that link segments with higher intensities are split more homogeneously on average.

Interestingly, Fig. 8 also shows that the simple strategy of distributing the rainfall rates homogeneously along the path of the links results in slightly better performance than the ensemble average of CLEAR. Nevertheless, they also systematically overestimate light rainfalls and underestimate heavy ones. This behavior is caused by averaging of extremes as indicated in Fig. 5.

### 3.4 CLEAR ensemble variance

The stochastic nature of CLEAR means that it can be used to generate ensembles of cascade realizations to assess the effect of measurement/model uncertainty on disaggregated rain rates. Figure 9 shows the containing ratio (*CR*) and average band width (*ABW*) conditional to the rain rate and CML length. Both metrics were evaluated over all 210 time steps. *ABW* increases with rain rate and CML length. This reflects what we expect, i.e., that the uncertainty of the disaggregation increases with increasing rain rates and CML path length. However, the *CR* values below 90 % indicate that the ensemble variance and hence

the uncertainty represented by the band width are underestimated. The underestimation is the largest for light rainfall with rain rates below 1 mm h$^{-1}$ (*CR* = 0.18 – 0.47). The best performance, although not optimal, is achieved for rain rates between 2 – 10 mm h$^{-1}$ (*CR* = 0.60 – 0.80). On average, shorter CMLs tend to have lower *CR* than longer CMLs. Higher *CR* for longer CMLs is probably related to wider bands (higher *ABW*) of longer CMLs caused by the systematic overestimation of modelled SD (5) when compared to empirical SD (Fig. 2).




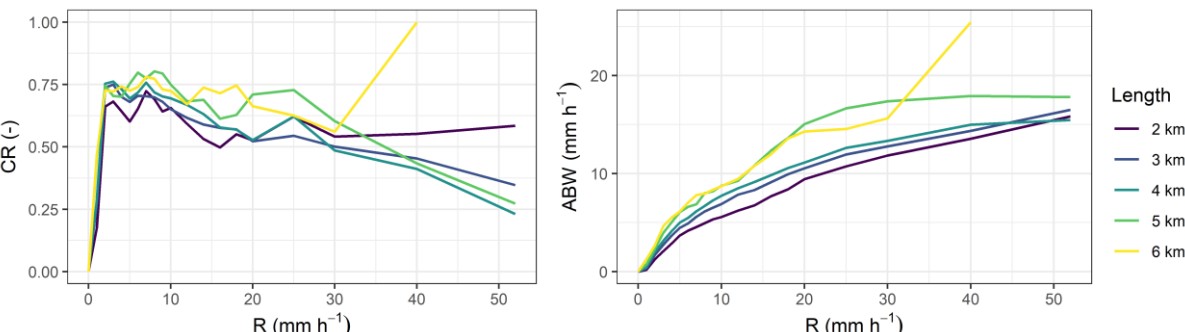

**Figure 9: Containing ratio (left) and average bandwidth (right) conditional to rain rate and CML length.**

## 4 Case study

### 4.1 CML data and rainfall retrieval

In this section, the strengths and weaknesses of CLEAR are demonstrated on real CML data during single heavy-rainfall event on 21[th] September 2014 with duration of two hours and average rainfall depth 18 mm. Original 10-s CML attenuation data were averaged over 5 min to match the temporal resolution of the weather radar data. The averaged CML attenuation data were then processed with a standard baseline and wet antenna identification methodology (see e.g. Chwala and Kunstmann, 2019). More specifically, the rainfall-induced attenuation along each CML was calculated by subtracting a constant baseline attenuation equal to the median of the total losses during September 2014. The wet antenna attenuation correction is a modified version of the Kharadly model (Kharadly and Ross, 2001) proposed by Pastorek et al., (2022b) with a single set of model parameters for all the CMLs. The parameters of the wet antenna model were optimized by minimizing the average squared difference between the path-averaged rain rates from the CMLs and the reference path-averaged rain rates obtained from gauge-adjusted weather radar. Rainfall-related path attenuation is converted to rain-rate using a standard power law model (2) with ITU parameters (ITU-R, 2005). Note that six CMLs experienced outages during this day, which means that only 71 out of the 77 CMLs were used.

Reference and CLEAR-disaggregated rain rates are resampled in the same manner as described in Sect. 2.5. The threshold for maximal segment length in the GMZ algorithm is set to 1 km to match the resolution of the reference weather radar product.

### 4.2 CLEAR performance for the case study

The performance of CLEAR is first demonstrated on the same set of CMLs as in Sect. 3.1. Figure 10 shows disaggregated rain rates obtained using CLEAR during time step 21. It shows that the CML path-averaged rain rates are systematically underestimated compared to the reference. Consequently, CLEAR also shows a systematic underestimation of the rain rates. However, this is not a shortcoming of the method but more an issue of the CML data themselves. The locations of the min/max rainfall intensities along the CMLs are better for CML 63 than for CML 9. Overall, CLEAR reproduces the min/max more




reliably than GMZ (Fig. 11). However, the results are significantly affected by the uncertainties in CML rain rate estimates.

Also, and although they are bias-corrected, the radar rainfall estimates are likely to be affected by local biases as well. The minima, maxima, and standard deviations are similar to the values obtained on the simulated data and most reliably estimated by averaging the statistics over the ensemble. CLEAR has a slightly better performance than GMZ when evaluating segment-by-segment matches between reference and disaggregated CML rain rates: The *RMSE* values are 3.00 mm h$^{-1}$ and 3.43 mm h$^{-1}$ for CLEAR and GMZ respectively and the $R^2$ is 0.38 and 0.30 respectively.

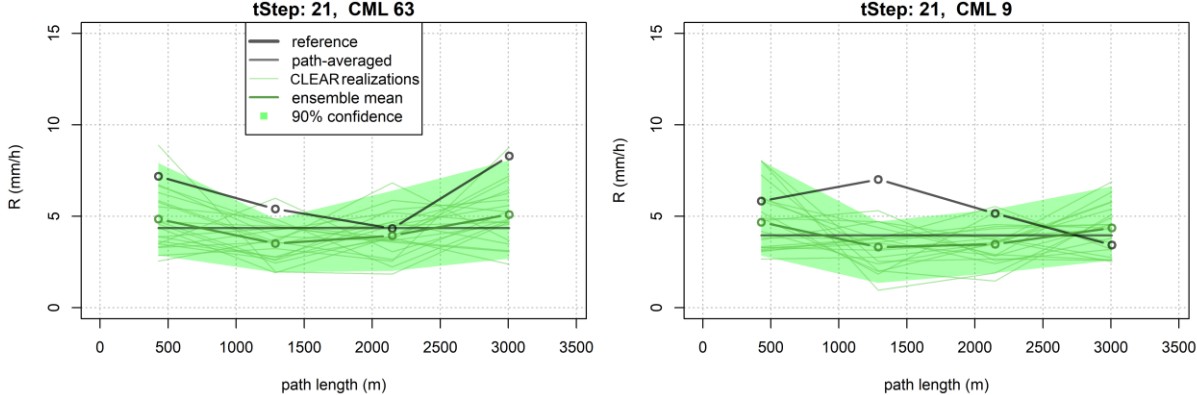

**Figure 10: CLEAR disaggregation of rain rates during one time step demonstrated on real data from two CMLs of similar path length.**


The results of the case study are strongly affected by a large discrepancy between the CML path-averaged rain rates and the

reference. First, a rainfall amount along CML path which is being disaggregated is determined by the estimated rain rate. Second, the under- or overestimation of initial rain rate affects the cascade generator model and estimated breakdown coefficients. Despite large uncertainties, CLEAR is still able to reproduce variability and extremes along a CML path more reliably than GMZ benchmark.

The discrepancy between reference and estimated rain rates is partly caused by inaccurate path-averaged CML rainfall

estimates, nevertheless, it is also related to the limited reliability of the radar rainfall product used at the 5-min temporal resolution. In general, a radar adjustment at high resolution is highly challenging e.g. due to the scale discrepancy of a radar pixel and a rain gauge catch area, possible displacement of rainfall field due to rainfall advection, etc. ( see e.g. Ochoa-Rodriguez et al., 2019; Schleiss et al., 2020).

Further evaluation on real data should focus on data aggregated over longer time intervals (e.g. 30 min or hourly data) for

which adjusted radar quantitative precipitation estimates are more accurate. As rain rate aggregation over longer intervals leads to the smoothing of local extremes, the effect of any disaggregation will be less pronounced. Next case study evaluating CLEAR on aggregated data should thus focus on a network outside of a city which is commonly characterized by longer CMLs.



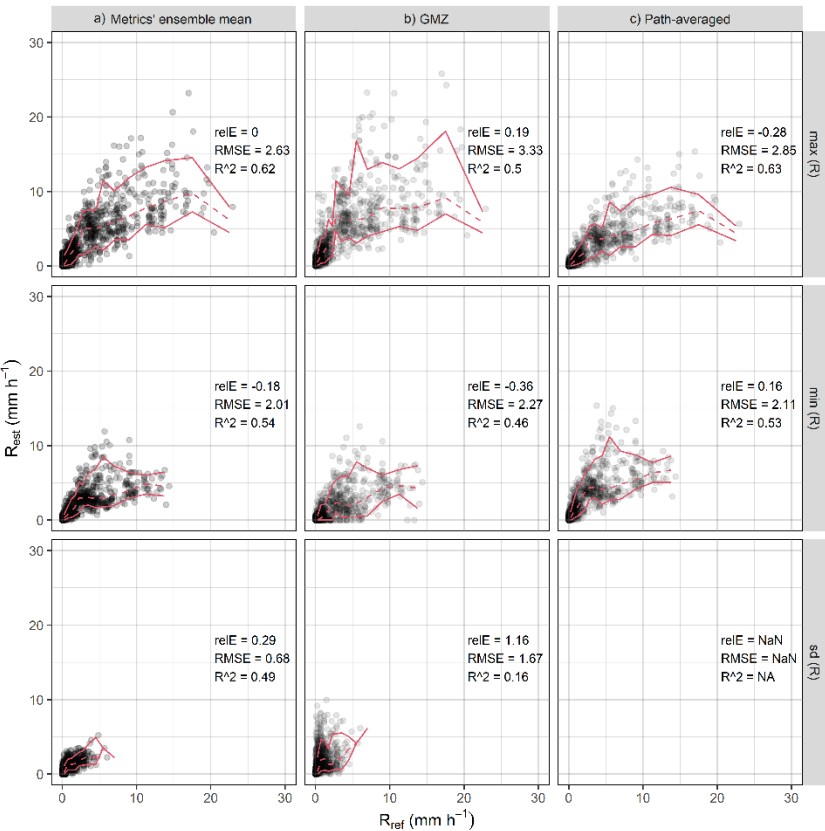

**Figure 11: Statistics of reference (radar) rain rate along a CML path for each time step and CML compared to the statistics when disaggregating real CML observations with: a) CLEAR algorithm and calculating ensemble mean of each metric, b) benchmark GMZ algorithm, or c) using path-averaged rain rates without any disaggregation. Diagonal dashed red lines indicate identity.**

## 5 Discussion

In this section we discuss results obtained from the experiment with simulated rainfall fields and identify factors influencing the performance of CLEAR. We also highlight advantages and limitations of CLEAR compared to the benchmark algorithm and explore potential research directions to address some of these limitations.

### 5.1 Modeling the standard deviation of the cascade generator

The logit-normal cascade generator model behind the CLEAR algorithm assumes a simple power-law relation between the standard deviation (SD) of the cascade weights, the path-averaged rainfall intensity and link length (Eq.,5). Using the simulated rainfall fields, we can study the actual standard deviation of the empirical breakdown coefficients for a large number of CMLs links and compare them to the modeled ones to see how well the generator fits the data. Figure 12 shows the empirical breakdown coefficients when evaluated for each of the three simulated rain events, together with the global, fitted power-law





model for the standard deviation (SD). It shows that the global SD model obtained by combining all the events together and imposing a power-law model significantly differs from the actual SD values for a given event. For starters, there are clear

differences in the magnitude of the SD (for a given rainfall intensity and link length) from one event to another. Also, the patterns can be quite different. For example in event 1, the SD tends to increase for rainfall intensities between 1 – 5 mm, which is very different from the gradual decrease with intensity predicted by the model. The same ups and downs can be observed for event 2 and may be the consequence of the non-stationarity of the generator model in space and time. The fact that our simple cascade generator model cannot accommodate such patterns could explain the conditional bias with rain rates

as well as the inability to adequately capture the location of min/max rainfall intensities along the link (Fig. 4 and Fig. 10).

To investigate this issue in more depth, we analyzed the empirical breakdown coefficients of the CML network in Prague using one full year of bias-adjusted radar data and many different rain events. We found that the magnitude of the SD also seems to be related the maximum rainfall rate in the domain, however, the incorporation of this behavior through an additional parameter created more problems than it solved and often led to overfitting. The non-stationarity issues and the superposition of different

generator models inside the domain was not explored, and further research is needed to understand how it could be detected and taken into account.

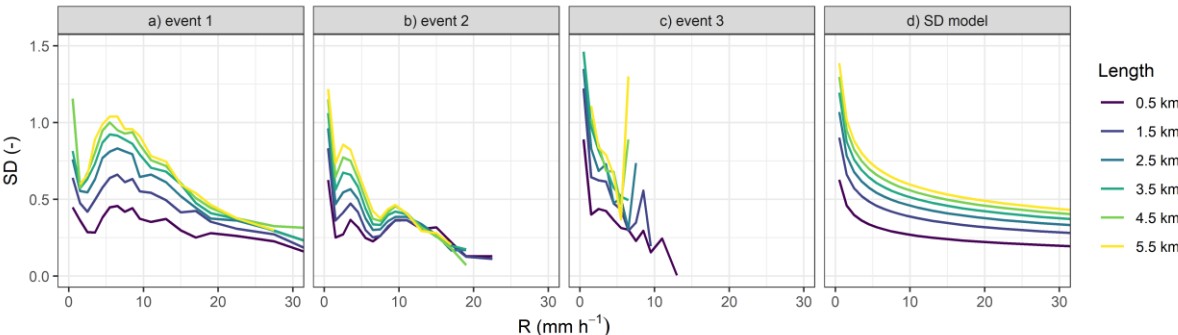

**Figure 12: Empirical (a-c) and modeled SD (d). SD of empirical breakdown coefficients is shown separately for three events evaluated in this study.**

## 400    5.2 Sensitivity of CLEAR disaggregation to the spatial coherence rule

CLEAR is efficient in estimating rainfall extremes and variability along CML path (Fig. 5 and 6) and, in this respect, clearly outperforms the GMZ algorithm and path-averaged rain rates. However, it also struggles to reliably predict the position of the smallest/largest rainfall rates along the link, as clearly demonstrated by the results of the segment-by-segment evaluation (3.3). To better understand where the errors in CLEAR originate from, we took a closer look at the performance of the spatial

coherence rule itself. Specifically, we performed two additional analyses with the high-resolution reference rainfall. In this simulation setting we could study how well the empirical splits based on interpolated rainfall rates from coarser scales actually are.





In the first analysis, we evaluated precision of the empirical spatial coherence rule, i.e. ratio of splits when link segment correctly received the shortest length and, therefore, the highest rainfall intensity along its path. On average, about 1/3 of the splits performed using the empirical rule were wrong leading to displacement of rainfall peaks and minima. In the second analysis, we applied CLEAR using an optimal spatial coherence rule based on the true rain rate along a CML path. We found that using an optimal spatial coherence rule significantly improves the performance of CLEAR on a segment-by-segment basis: For example, the *RMSE* decreased from 2.95 to 2.15 mm h$^{-1}$ and $R^2$ increased from 0.75 to 0.86. However, the optimal spatial coherence rule had almost no effect on the average performance statistics for the min/max and standard deviation of rain rates along a CML. The shortcomings of the empirical spatial coherence rule are thus not critical for applications where statistical distribution of rain rates is more important than their exact positioning. For example, for improving rainfall estimation from CMLs at lower (X, Ku band) or higher frequencies (E-band, W-band), where attenuation- rainfall relations can be significantly nonlinear and thus a commonly used  assumption of uniform rain rate along a CML path potentially leads to higher errors.

## 5.3 Ensemble variance

Ensemble variance arises from a stochastic nature of the cascade generator (5), however, the relation between the two is not straightforward. For example, the ensemble variance in CLEAR may be strongly affected by the spatial coherence rule used to split the segments. Our analyses show that different realizations of CLEAR disaggregation preserve similar rainfall pattern along a CML path, R-squared between the realizations is 0.84-0.86. In general, the uncertainty estimates derived from the CLEAR ensembles tend to be underestimated; the mismatch is highest during light rainfall (0 - 1 mm h$^{-1}$) and heavy rainfall ($R > 30$ mm h$^{-1}$) (Fig. 9). Additional analyses (not shown) suggest that for light rainfalls, the underestimation may be due to the difficulty in reproducing rainfall intermittency and reparameterization of SD model discussed in Sect. 5.1 could help in this regard. For heavy rainfall, the estimated variability and position of rainfall peaks along the CMLs tend to be incorrectly estimated (Fig. 4, right). To improve, it might be necessary to design better, more elaborate and spatially variable coherence rules (e.g., as a function of CML density) to account for the uncertainty related to spatial coherence. Alternatively, some randomness could be introduced in the splitting rule. For example, by randomly re-assigning the peak rainfall rate to the other side of the CML with a frequency of about 1/3 or less (especially at the first cascade levels). More realistic ensemble spread might also be achieved by improving the cascade generator model. For example, by locally adapting the spread of the cascade weights to account for the spatial correlation structure of the rain and other non-stationarities (e.g., proximity to dry areas).

## 6 Conclusions

A new disaggregation algorithm for CMLs named CLEAR (CML segments with equal amounts of rain) has been proposed. Within CLEAR, the splitting of link segments is controlled by a multiplicative random cascade generator, whose standard deviation depends on the rain rate along the CML segment and the length of the segment. Rain rate information from neighbouring CMLs is used to estimate the areas of largest/smallest rainfall intensities and thus preserve spatial consistency





during the splitting. The stochastic character of CLEAR makes it possible to represent uncertainty as an ensemble of rain rate
distributions along a CML.

Evaluation of the CLEAR algorithm on virtual rainfall fields shows good overall performance and realistic spatial patterns. CLEAR outperformed the GMZ benchmark both in the simulations and on real data. The case study, however, revealed challenges related to uncertainties in CML quantitative precipitation estimates, which are common to all methods. Despite the encouraging results, lots of potential for improvement remains. For example, the ensembles generated by CLEAR still
significantly underestimates overall variability along a CML path. The segment-by-segment evaluation also shows that performance is negatively affected by errors in positioning of rainfall extremes along the CML. A better spatial coherence rule, accounting for rainfall advection and the introduction of more randomness into the splitting rule could help in this regard.

In conclusion, CLEAR can help in generating more representative rainfall distributions along CMLs, which is important for the spatial reconstruction of rainfall fields from path-integrated CML data. However, further research is needed to improve the
spatial coherence rule and cascade generator model. CLEAR might also help to model rainfall intermittency along a CML path, albeit, this feature needs to be investigated in more detail probably. Future work should investigate how to deal with non-stationarity of the rainfall field and the cascade generator model, and how to incorporate data from previous time steps. It might also be interesting to investigate the performance of CLEAR when applied to CMLs at lower (X, Ku band) or higher frequencies (E-band, W-band), where attenuation- rainfall relations can be significantly nonlinear and thus an assumption of
uniform rain rate along a CML path potentially leads to higher errors.

**Appendix A**

The spatial coherence rule is evaluated using gridded rainfall fields, which are reconstructed from CML data at each cascade level: initially from the original path-averaged rain rates and at further cascade levels from disaggregated rain rates along disaggregated CML segments. The initial resolution is 4 x 4 km$^2$ and this resolution is refined after each cascade level such
that the new grid size resolution is the original size divided by the order of the cascade level. The initial and refined resolution approximately correspond to the length scales of the longest CMLs resp. their segments evolving as the result of the disaggregation.

Rainfall fields are constructed as follows: First CML segments are assigned to the grid cells by evaluating the overlap between the cells and the midpoints of the segments. Then, the cell rain rate is estimated as the average rain rate of the CML segments
belonging to the cell. The cells that do not containing any segment are marked as cells with not available rain rate and are omitted from the evaluation of the spatial coherence rule.

Rain rates for the purpose of evaluating the spatial consistency rule are sampled from the rainfall field at 24 regularly spaced positions near each end node of a CML segment. The distance between these positions is set to 1/3 of the grid size of the CML-based rainfall field. The splitting example is shown in Figure A1.





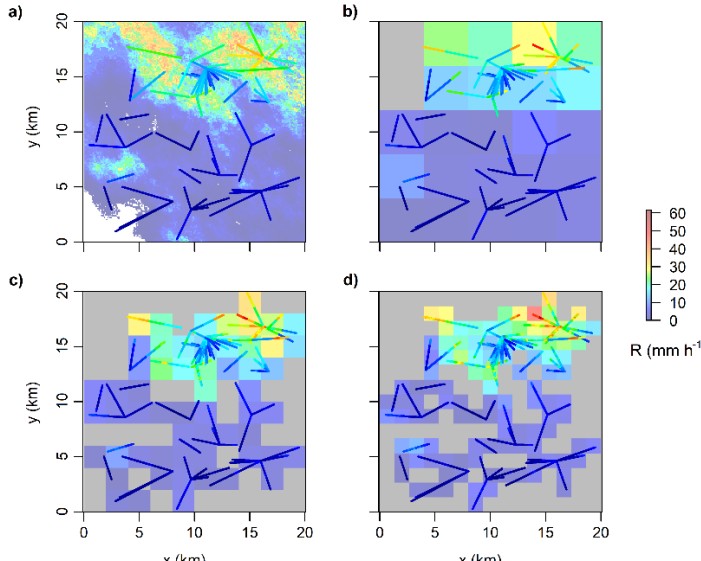


**Figure A1: a) Reference rainfall field and CMLs with color-coded path-averaged rain rates. b-d) Gridded rainfall used to evaluate spatial coherence rule when splitting CML segments at successive levels of the cascade. Grey color indicate cells with not available rain rate values.**

## Appendix B – Fitting of SD model

The SD model is fitted to sample SD estimates of empirical breakdown coefficients $W_0$ obtained from synthetic CML networks (Sect. 2.7). To ensure zero mean, the original breakdown coefficient $W_0$ and the difference $1 – W_0$ are merged to one population: $W = W_0 \frown (1 - W_0)$. The breakdown coefficients $W$ are then transformed to a Gaussian using the left-hand side of (3) and grouped according to path lengths and path-averaged rain rates of the parent links that generated them. The length classes are equidistantly spaced with a bin size of 500 m between 500 and 6500 m. The rain rate bin sizes are 1 mm h$^{-1}$ between 0 – 10

mm h$^{-1}$, 2 mm h$^{-1}$ between 10 – 20 mm h$^{-1}$, and 5 mm h$^{-1}$ between 20 – 55 mm h$^{-1}$. The decreasing size of the bins with growing rain rates reflect naturally lower representation of higher rain rates in the population of CML rain rates. Sample SD values are calculated for each group having at least 50 samples in them.

   SD model (5) is optimized using the simplex method for function minimization (Nelder and Mead, 1965) implemented in the `optim()` function available within the statistical computing language R (R Core Team, 2020). A following objective

function is minimized:

$$L = \sum log \left( \left| \left( SD_{sample} - SD_{model} \right) \right| \right), \tag{7}$$

Where $SD_{sample}$ is sample SD and $SD_{model}$ is modelled SD. The optimization is performed for sample SD obtained for breakdown coefficients of all synthetic CMLs during all three virtual rainfall events (Sect. 2.5).




*Data and code availability*. The essential data and codes underlying this work are publicly available at 4TU Research Data repository (Fencl and Schleiss, 2025). CML attenuation data for the case study are subjected to non-disclosure agreement (NDA) and can be shared on request to the corresponding author under conditions defined by the NDA. Similarly weather radar data and rain-gauge can be shared on request to the corresponding author under conditions defined by the a license
agreement.

*Author contributions*. MF performed the analysis, prepared all the figures and wrote the first draft. MS provided advice, gave critical feedback on previous draft versions and helped improve the quality of the writing. MF and MS together prepared, annotated, and published the dataset including the code used for CLEAR disaggregation.


*Competing interests*. The authors declare that they have no conflict of interest.

*Acknowledgements*. The authors greatly acknowledge financial support from the Czech Science Foundation (GACR) project MERGOSAT 24-13677L. We would like to thank T-Mobile Czech Republic a.s. for providing the CML data. Special thanks
are extended to Pražská vodohospodářská společnost a.s. for providing rainfall data from their rain-gauge network and Pražské vodovody a kanalizace, a.s., who carefully maintains the rain gauges. Finally, we would like to thank 4TU Research Data for facilitating the publication process of our codes and data under the FAIR principles.

*Financial support*. This research has been supported by the Grantová Agentura Ceské Republiky (grant no. 24-13677L).

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
