# Peer review of "CLEAR: a new discrete multiplicative random cascade model for disaggregating path-integrated rainfall estimates from commercial microwave links"

_EGUsphere, 2025_

## Author Response (AR1)

*We thank both reviewers for their valuable comments. While we have addressed each reviewer's feedback individually, we have also taken into account overlapping concerns and addressed them comprehensively in our responses.*

Reviewer 1
This paper suggests a novel approach to disaggregate path-average rain rate over CML. The topic is relevant and interesting for the community. The paper is very "clear" ;-) to read. Indeed the methodology is presented in a straightforward way, results displayed and discussed in convincing way, and current limits are also properly addressed. In general, I think that the paper only require minor modifications before it is suitable for publication.

Minor specific comments:

- l. 97: "L1" should be in italic.

*Corrected.*

- l. 110-114: it is indeed one approach, but there are others. Did authors carried out some tests to opt for this approach ? This should be clarified.

*We have designed the rule following main principles described in Schleiss 2020, from whom we also translated 2D equal-volume-area disaggregation to 1D CLEAR disaggregation. We discuss limitations of our SC rule design in the Discussion (Section 5.2 of the original manuscript). While our experiments using an optimal SC rule (derived from simulated reference data) suggest that further improvement in peak positioning may be possible, these gains are unlikely to significantly affect average performance metrics such as the min/max or standard deviation of rain rates along a CML.*

- Section 2.6: maybe say few words on the uncertainty associated to radar QPE. Is it really a reliable reference ? (few things are said l. 345 on the topic)

*Agreed, we have included a brief evaluation in the paragraph about the radar adjustment of radar accuracy based on cross-validation against rain-gauge data.*

- l. 176-177: could authors clarify why does the parameters of the cascade generator depend on link orientation and length ?

*The splitting is related to the rainfall spatial correlation structure. Longer links are more likely to have variable rainfall along their path, e.g. part of the link path can be inside a convective cell and part not affected by rainfall at all. Thus representation of different link lengths is important. Furthermore, the link orientation might affect splitting due to rainfall anisotropy. We have extended the sentence (originally on line 176-177) to mention this.*

- l. 180: the whole point is in "realistically"… How sure are you that your rainfall simulations are "realistic" ? I also agree that the other approach also has intrinsic limitations as pointed out by the authors. Maybe carrying out the calibration with both method and highlighting the differences would be more convincing.

*We have added a paragraph explaining how the fields were generated and what we mean by "realistic structure." In our response to Reviewer 2, we also included additional details on the calibration of the SD model and the sensitivity of the results. Given the remarkable robustness of the results with respect to the chosen SD model, we believe that there is little value in introducing and discussing an alternative calibration approach in the paper.*

- Fig. 3 (right): It would be interesting to find a way to always display the differences according to rain rate.

*We find scatterplot convenient for displaying correspondence between empirical and modeled SD. We have, however, modified the right plot such that different rain rates are now indicated by point shapes.*

- Section 2.8: please justify better the choice of GMZ as benchmark. It seems as a simplistic approach with regards to what is suggested in this paper, hence not surprisingly it performs worse.

*According to our review of the literature, GMZ is the only (simple) rainfall reconstruction algorithm which explicitly reconstructs rain rates along the CML path. More complex algorithms, e.g. tomographic reconstruction methods (Cuccoli et al., 2013; Giuli et al., 1991; Zinevich et al., 2008) or random mixing (Haese et al., 2017) exist but they require careful parameter tuning and/or the selection of an underlying rainfall model. While these methods may outperform simpler ones under specific conditions and with well-chosen parameters, we believe they are not practical to use as a general benchmark. Note that the advantages and limitations of different algorithms are discussed in the introduction.*

- L. 245 – 248: I understand the point, but it would be interesting to show quantitatively the increase in performance in areas with denser CML concentration.

*The quantitative evaluation of CMLs with respect to their position within the network is actually presented on Figure 7 in the Section 3.2.*

- Section 5.1: wouldn't it be possible to use empirical curves for SD as a function of rain rate instead of a modelled one

*Indeed, the use of empirical SD curves, estimated separately for each event (or even for each field) would likely result in better performances. However, this requires additional independent rainfall measurements (e.g., from a radar), which may not be available everywhere. In this paper, we wanted to develop a method that can be applied to cases where only CML data are available.*

Reviewer 2
The authors present a new method to distribute mean rainfall rates from a commercial microwave link (CML) network along the CML line of sight. The underlying random cascade model considers spatial coherence within the CML network and provides

uncertainty estimates. Required model parameters that determine the uniformity of the rainfall distribution along each transect are fitted to three virtually generated rainfall fields and the Prague CML network. Finally, the method is compared to the method by Goldshtein et al. (2009) for the same three virtually generated rainfall fields and radar observations of a single two-hourly rainfall event in Prague on 21 September 2014. The results show improved reconstruction capabilities by the proposed method.

While the newly developed method might be beneficial compared to existing methods, the performance evaluation presented in this work is not convincing. Below are several general and specific comments that the authors must address in a revised manuscript.

Authors:

*First of all, we would like to thank the reviewer for their thorough assessment of our manuscript. We acknowledge several of the concerns and have thought about how to address them in the revised manuscript (see major comments below).*

*However, we first would like to clarify and elaborate on the concerns regarding the limited extent of the dataset used to evaluate the performance of CLEAR. While this manuscript includes a comparison with an existing method, it is not intended as a comprehensive comparative study. The core contribution of our paper lies in the formulation of a new, spatially coherent random cascade model for disaggregating path-averaged rainfall from commercial microwave link (CML) networks. The evaluation using 210 virtual rainfall fields, along with the application to real-world data, primarily aims to illustrate the capabilities and limitations of the new CLEAR disaggregation algorithm and to demonstrate its potential for further development.*

*We acknowledge that broader validation using larger and more diverse datasets would be a natural next step (in a follow-up paper), and we will make this clearer in the revised paper. However, we believe the current evaluation is appropriate and sufficient for a paper that focuses on methodological development.*

General comments

1. Sample size

This work is based on a tiny sample size of just four rainfall events: 3.5 hours of virtually generated rainfall fields and 2 hours of radar observations in Prague. This sample size does not provide a statistical basis for model evaluation and intercomparison with the Goldshtein method. I suggest using at least a full annual cycle of weather radar and CML data to capture a larger number of heavy precipitation events. This will allow for a more differentiated comparison with the Goldshtein method under different seasons and precipitation regimes. Also, it provides insights into whether the proposed method is even valid for an entire season or if the underlying parameters of the SD model need to be adjusted for convective/stratiform types.

*We understand the reviewer's concern regarding the extent of testing and agree that comprehensive validation across a wider range of conditions is crucial for future development. However, the primary contribution of this paper is not a performance evaluation, but the demonstration of a novel methodological approach. Our simulation experiments are designed to show the feasibility of the method and to illustrate its potential strengths and limitations through targeted examples. The core analysis is based on 210 rainfall fields from three distinct events, producing over 16,000 observations from 77 CMLs, with rain rates up to 50 mm/h and path lengths between 1.5–5.5 km. While the case studies are not fully representative of all possible real-world scenarios, we believe they are sufficient to build intuition, reveal the algorithm's general behavior, and identify key challenges, thereby laying the groundwork for more extensive future evaluations. We will clarify this distinction more explicitly in the revised manuscript to avoid potential misunderstandings.*

*The implemented revisions can be found in lines 68–71, 447–449 and 547–549 of the revised manuscript (version with tracked changes).*

**2. Virtual rainfall fields**

This work mainly relies on virtual rainfall fields generated following the method by Schleiss et al. (2012). Most readers probably won't be familiar with this method. Therefore, I suggest adding a short description of how the method works and which parameters are required to generate rainfall fields. It should be discussed if any assumptions used to generate the rainfall fields are similar to assumptions of the proposed CML disaggregation method. Also, the spatiotemporal structure is described as "realistic," but no evaluation against any observations, e.g., radar, is performed. Especially, spatial features between 100 to 1000 m might lack any reference as those scales are typically below the horizontal resolution of weather radars.

*Thank you for this suggestion, we have added a paragraph into the section Simulated rainfall fields (2.1 in the revised manuscript) describing in more detail their main features and the method used for their generation.*

**3. Model evaluation**

There is a lack of independent rainfall fields for the intercomparison of the proposed method with the Goldshtein method. This work fits the parameters of the SD model to the virtual rainfall fields and then uses the same rainfall fields for the model intercomparison. Therefore, it is no surprise that the proposed method outperforms the Goldshtein method, e.g., the RMSE of capturing the maximum rainfall is about 70% lower (Fig. 5). The results for the independent two-hourly radar field show similar results as the Goldshtein method and path-averaged rainfall. I suggest using independent radar observations for evaluation over a long time period (not just two hours on 21 September 2014). Moreover, I was wondering if high-resolution atmospheric models (large-eddy simulations) could provide a source of rainfall fields that could be used instead of the statistically generated rainfall fields.

The SD model was indeed fitted to all 210 rainfall fields, meaning the training and testing data were not fully independent. However, we believe that for illustrating the capabilities of CLEAR, this remains a reasonable approach, especially since we also compare the globally fitted model to event-specific SD models.

That said, we took this concern seriously and conducted additional tests using an entirely independent dataset of 2'713 unadjusted radar observations from the Czech Republic (1 km x 1 km, 5-minute resolution) between April and October 2014.

Figure R1 compares the original SD model (RMSE = 0.45, rel. error = 0.16) with the new model based on independent data (RMSE = 0.83, rel. error = 0.35). The new model underestimates SD for short CMLs, overestimates it for longer ones, and tends to underestimate rain rates below 1 mm/h. The fit is worse, the parameters are clearly different and overall, the new model tends to overestimate the spread compared to the one we used in the paper.

Despite the substantial differences between the two SD models, the performance of the CLEAR algorithm remains virtually unchanged (Figure R2), highlighting its strong robustness to the choice of SD model. This property was also noted by Schleiss (2020) for the EVA cascade model. The explanation lies in the nature of the cascade process and functional form for the cascade generator model: although long CML segments may split quite differently at the initial levels, where SD values are higher, these differences quickly reduce in the later stages of the cascade. As the SD values decrease rapidly with each iteration, the resulting subdivisions become increasingly uniform, making the final disaggregation less sensitive to the specific SD model used.

[Figure]

*Fig R1: top row: Empirical SD from 210 fields,SD model fitted to empirical SD, and comparison of empirical SD. bottom row: modeled SD obtained from CZ radar data, comparison of empirical SD form 210 fields and SD modeled from CZ radar data.*

[Figure]

*Fig R2: one realization of the CLEARalgorithm with the original SD model fitted to 210 fields (top row) and results obtained with the SD model fitted on independent CZ radar data (1 km$^2$ resolution).*

*The robustness is now discussed in more detail in the discussion section, specifically subsection 5.1.*

**4. Structure**

I strongly encourage the authors to revise the structure of this paper from chapters 2-5:

- Split "Material and methods" into separate "Data" and "Methods" sections. This will avoid the current jumps between methodology and data sections.

- Move the methodology details from the appendix to the respective location in the method section. I do not see a reason for describing these steps in the appendix.

- Merge the two result sections ("results" and "case study") or rename the "results" section to avoid confusion on where to find all results.

- Avoid presenting new results in the discussion section.

*Thank you for these suggestions. We have split the material and data section and merged the results section with the case study. This required also rewriting some sentences in the methods and data sections. We, however, respectfully disagree with the final comment about not introducing new results in the discussion part. The analyses included in the discussion section go beyond the core evaluation but contribute meaningfully to understanding and interpreting the main findings. In our view, the boundary between presenting results and contextualizing them through supporting analyses is not always sharply defined and often reflects the author's stylistic choice. These additional analyses are not central to the evaluation but allow us to provide more insight into potential limitations and the robustness of our approach, which aligns with common practice in methodological studies.*

*Similarly, we do not consider the content presented in the appendixes central to the methodology but still important to include for the sake of reproducibility. The Appendix section is in our opinion appropriate for such content.*

**Specific comments**

line 46: Word missing: "distribution of"

*Corrected*

line 53: Add a new line before "This paper addresses"

*Agreed*

line 58-59: Provide references.

*We now refer readers to Molnar and Burlando (2025) - https://doi.org/10.1016/j.atmosres.2004.10.024.*

line 60: This sentence is incomplete. Add information on the reference CLEAR gets compared to.

*We had in mind the above mentioned algorithms. We replaced 'better' with 'well'.*

line 77: This should be mentioned in the code and data availability section and not here.

*Agreed.*

Equation 4: The equation is not correct. Dividing by R0 and knowing that R1 = R2 (line 96) leads to W1 = W2.

*This is a misunderstanding. On line 96, we talk about equal rainfall amounts (rain rates integrated over a path), whereas $R_1$ and $R_2$ are point rain rates. Rainfall amounts are obtained when the rain rate is integrated over the link (or link segment) path. To clarify this we have added units after rainfall amount and length $L_0$ (km). "... containing half of the original rainfall amount (mm $h^{-1}$ km). The ratio between the parent length $L_0$ (km) and resulting segment lengths…"*

line 104: Repeats line 97

*We will modify the sentence on line 97 to: "The ratio between the parent length $L_0$ and resulting segment lengths $L_1$ and $L_2$ is determined by drawing random weights W from a cascade generator model with logit normal probability distribution:"*

line 459: How does the choice of 4km x 4km initial grid size affect the final prediction? Is there a way to compute this initial grid size for a given CML network?

*We did not study this aspect in detail, however, the initial grid size should reflect length scales over which CMLs integrate and the density of a network. The size of the 4 x 4 km grid was a pragmatic choice which enabled us to construct a field covering the entire domain without need for interpolation.*

*The analysis with true spatial coherence (SC) rule shows that the SC rule affects the results significantly, however, its further improvement is beyond the scope of the manuscript.*

line 465: "contain"

*Corrected, thank you!*

line 467: I do not understand this paragraph and what is meant by "24 regularly spaced positions". Please rewrite this and clarify.

*We have rewritten the paragraph and added into Figure (A1) an example of the SC rule sampling scheme.*

line 118: How does the choice of the threshold affect the predictions, and which value is used here?

*The threshold, i.e. rainfall amount at which the splitting terminates, is set to 1 mm $h^{-1}$ km We have added this information into the revised manuscript: "In this analysis, the threshold is set to 1 mm $h^{-1}$ km, which corresponds approximately to the attenuation of 1/3 dB by the 23-38 GHz CMLs, i.e. the value matching the quantization of CMLs employed in the case study."*

*The choice of this threshold may influence how intermittency is represented, however, we have not explicitly tested its impact. In contrast, choice of the threshold value has likely a small effect on the estimated rainfall maxima due to the resampling approach used in this study.*

line 124: References to specific scripts are not very useful for the reader. If those are relevant to understanding this paper, I suggest pseudocode.

*Agreed, we have removed references to the specific scripts.*

line 127: Provide a more meaningful section header instead of "Approach".

*This section has been removed and its content distributed in appropriate places in the data and method sections.*

line 128: This CML data has to be introduced and described first.

*The new structure of the manuscript reflects this concern.*

line 131: GMZ not introduced.

*"Ditto"*

line 151: Move this to the "code and data availability" section.

*Agreed*

line 157: What is meant by "the case studies"? I don't think any case studies (time, place) were introduced yet. Also, which time period does the radar data cover? Mention this in the text with basic statistics on the rainfall amount, maximum rain rates, and number of rain events.

*The case study is now first mentioned already in the data section, where the datasets used for the case study are introduced.*

line 180: How was it ensured that the simulated rainfall fields correspond to the "local climatology"? And for which place were they tailored to?

*Note that the statement in this sentence has a general meaning—namely, that simulated rainfall fields which realistically represent local climatology ensure that the SD model remains consistent with that climatology.*

*The general approach for aligning simulated rainfall fields with local climatology is now described in Section 2.2, where the fields are introduced. In this particular case, the fields were tailored to the conditions in Lausanne, Switzerland. Nevertheless, as already explained in the response to the general comment, this setup adequately serves the purpose of the evaluation, even though the study area is located in Prague, Czech Republic.*

line 188: There is no "=" sign in equation 3 and thus no left hand side, please check.

*Corrected, thank you for spotting this typo.*

line 191: See comment on line 124.

*The reference has been removed.*

Figure 2: Typo in the caption: "sum"

*Corrected.*

Figure 3: What causes the "wave" pattern between 0-10 mm/h? Could radar data provide a way to verify this empirical relationship, at least for the longer CML links?

*The wave pattern arises from the SD estimated during event 1 (Figure 12) and it is likely attributed to the highly intermittent nature of this event (see Table 1).*

*Radar data are indeed useful for studying empirical breakdown coefficients and their SD. When investigating different patterns with CZ radar data on the event basis, we saw many different patterns including wave patterns. However, when we evaluated the full*

*radar dataset, the relation between SD and rain rate converged towards exponential shape with no clear wave pattern (Figure R1). This is the reason why we in the end stick to the power-law model, despite its inability to reproduce the wave pattern.*

line 201: Are these the same rainfall fields that were used to fit the SD model?

*Yes. The reasons for using all the fields are explained in the response to the general comment no 3.*

line 232: Remove "(and in general)". While extreme events are rare in reality, it should be ensured that their number is sufficient for the evaluation.

*Removed*

line 246: How do I see if the CML link in that figure is isolated? Ideally, indicate those links in the map.

*We have indicated these CMLs in the map (Figure 1) and modified the figure title accordingly.*

Figure 5: Is this strong improvement in capturing the rain rate maxima expected? While the reference algorithm achieves an RMSE of about 6, the method presented here achieves an RMSE of 2.7, i.e., more than 50% reduction in RMSE. This is not discussed sufficiently in the text and I am afraid that this is due to the fact that SD parameters were fitted to the same rainfall fields.

*The GMZ model redistributes rain rates along a CML path based on rainfall information obtained from neighboring CMLs and the variability of rainfall along a path is not constrained by the CML path length or rain rate. In contrast, CLEAR constrains the redistribution by both length and rain rate through the SD model. The better performance of CLEAR was not so surprising for us, as the constraining implemented in CLEAR is well established in the disaggregation of 2D fields by multiplicative cascade models. We were, nevertheless, pleased to see that the improvement is substantial.*

*The SD model calibration and its effect on the performance of the CLEAR algorithm is in detail discussed in the response to the general comment 3 and in the revised manuscript also in Section 5.1. The additional results presented in the response show that the model is robust and relatively insensitive to a calibration dataset. The improved performance is therefore attributed to the inherent ability of multiplicative random cascade models to leverage the scale-invariant characteristics of rainfall during the disaggregation process.*

Figure 8: Highlight the point density as color shading to improve the readability of this plot.

*Thank you for this suggestion, we experimented with a hexbin plot, nevertheless, we found that a standard scatter plot enhanced with quantile lines better serves our*

*purpose. It more clearly illustrates both the data spread and the tendency to underestimate higher rainfall rates.*

line 311: Please define measurement and model uncertainty to avoid confusion. I would not use them interchangeably as the measurement uncertainty is typically associated with the input (path-averaged rain rate) and the model uncertainty with the model parameter uncertainty (SD model, coherence rule, ...).

*Indeed, the term model uncertainty fits here better.*

line 323: Move this section to data and methods. There is no reason for the separate sections "2.4 CML data" and "4.1 CML data and rainfall retrieval". Also, mention the exact time of the rainfall event in UTC.

line 340: Provide the actual time instead of "time step 21".

*Agreed, we have added time into the figure title and in the text.*

line 343: Rewrite the sentence and describe what "locations are better" means.

*Reformulated to: For CML 63, CLEAR accurately reproduces the distribution of higher rain rates at the end nodes and lower ones in the middle. In contrast, for CML 9, it does not adequately capture the peak located in the middle section of the path.*

line 344: It is true that, in this case, CLEAR seems to perform better than GMZ. However, even the path-averaged product provides better minima and maxima in both RMSE and R^2 compared to GMZ. And the R^2 of the minima and maxima from the path-averaged product is equal to CLEAR. While the sample size of just two hours of radar data is tiny, one would still expect to see similar trends as in Fig. 5 for the simulated rainfall fields.

*The evaluation in this case is highly affected by uncertainties of gauge-adjusted radar at 5 min resolution but also uncertainties in rain rates estimated from real CML data. Furthermore, aggregation of CML rain rates to 5 min leads to averaging of peaks and the effect of disaggregation is then smaller. We discuss these issues in the section presenting the case study results (Sect. 4.5).*

Figure 11: The transparency of points in column a is higher than in the other columns. All three data sets should be presented the same way for a fair comparison.

*Thank you for spotting this, we have corrected the figure.*

Figure 12: What is the difference between this figure and Figure 3, except that the empirical SD is shown per event? I would suggest combining both figures to show that, e.g., event 2 causes the "wave" pattern from 0-10 mm/h.

*Figure 3 illustrates the differences between the empirical and fitted standard deviations, while Figure 12 highlights how the empirical standard deviation varies across individual*

*events. In our view, presenting these figures separately better supports their respective purposes. Figure 3 shows the origin of the standard deviation model and how well it fits the data, whereas Figure 12 emphasizes potential inter-event variability that may not be fully captured by the model. Introducing the model's limitations prior to its evaluation could potentially confuse readers.*

line 391: I assume this analysis is not shown here? Using a year of radar data is a much more promising approach than the approach in this paper. I am surprised that overfitting occurred for an entire year of data; please specify.

*Indeed, this analysis is not shown here, however, as discussed in the response to comment about "wave pattern", the analysis provided us an insight into interevent variability of SD. Note also, that winter season was not included in this analysis and we now specify this in the dataset section.*

*The rationale for considering our evaluation approach appropriate for a methodological paper is discussed in the general comments. One of the key advantages we aimed to leverage is the sub-kilometre resolution of the simulated rainfall fields.*

*The overfitting was related to increased model complexity when an additional parameter was included and it is not necessarily related to the dataset size. Since we were unsuccessful when improving the SD model, we decided not to include the analysis into the core analyses presented in the results.*

line 394: Explain the "non-stationarity issues".

In revised manuscript we are more specific and use instead of "non-stationarity issues" "spatial non-stationarity of rainfall features over the domain"

line 405: The discussion chapter should not contain "two additional analysis". Instead, it should interpret, explain, and contextualize the results presented before.

*Thank you for this suggestion. However, as already explained in the response to the fourth general comment, we respectfully disagree on this point. The analyses included in the discussion section go beyond the core evaluation but contribute meaningfully to understanding and interpreting the main findings. In our view, the boundary between presenting results and contextualizing them through supporting analyses is not always sharply defined and often reflects the author's stylistic choice. These additional analyses are not central to the evaluation but serve to deepen insight into potential limitations and the robustness of our approach, which aligns with common practice in methodological studies.*

---

## Author Response (AR2)

We thank both reviewers for evaluating our revised manuscript. The first reviewer accepted the manuscript as is, while the second reviewer suggested three minor revisions, all of which we have addressed:

- We removed the statement from the abstract claiming that CLEAR outperforms a benchmark algorithm.

- We introduced the GMZ acronym (Goldshtein-Messer-Zinevich) at its first occurrence.

- We introduced the abbreviation SC rule for the spatial coherence rule.

In addition, we carefully re-read the manuscript and corrected several typos throughout. We also took the liberty of implementing a few minor technical revisions, listed below (line numbers refer to the final revised version with tracked changes):

- **L94**: Introduced the abbreviation DSD for *drop size distribution*.

- **L155**: Added units to the coefficients: $\alpha$ (mm h$^{-1}$ dB$^{-\beta}$ km$^{\beta}$) and $\beta$ (-)

- **L274–275**: Revised the introductory sentences to the *Results* section to reflect that it includes in the revised version also results from the case study.

- **L332 (Figure 7 title)**: Corrected the figure title for the right panel to indicate it contains path-averaged rain rates.

- **L403 (Figure 11 title)**: Updated the title to clarify that the red lines depict the median and quantiles, not the identity line.

- **L535**: Removed a reference to Section 2.5, as this section no longer exists in the revised manuscript.